A comparative study of root canal shaping using protaper universal and protaper next rotary files in preclinical dental education

Çelik Gül gulcelik@sdu.edu.tr gulcelik2016@gmail.com 1
Özdemir Kısacık Feyza 2
Yılmaz Emir Faruk 3
Mersinlioğlu Arife 4
Ertuğrul İhsan Furkan 5
Orhan Hikmet 6
1 Faculty of Dentistry, Endodontics, Suleyman Demirel University , Isparta , Turkey
2 Vefa Oral and Tooth Health Center , Afyonkarahisar , Turkey
3 Dentplus Oral and Dental Health Center , Bursa , Turkey
4 Araklı Bayram Halil State Hospital , Trabzon , Turkey
5 Faculty of Dentistry, Departmant of Endodontics, Pamukkale University , Denizli , Turkey
6 Faculty of Medicine, Biostatistics and Medical Informatics, Suleyman Demirel University , Isparta , Turkey
Hassan Ali
Electronic publication date: 2019 Aug 19
Publication date: 2019
Volume: 7
Electronic Location ID: e7419
Received 2018 Dec 19; Accepted 2019 Jul 5
Copyright: ©2019 Çelik et al.
Copyright year: 2019
Copyright holder: Çelik et al.
License: This is an open access article distributed under the terms of the Creative Commons Attribution License, which permits unrestricted use, distribution, reproduction and adaptation in any medium and for any purpose provided that it is properly attributed. For attribution, the original author(s), title, publication source (PeerJ) and either DOI or URL of the article must be cited.
License URL: https://creativecommons.org/licenses/by/4.0/

Keywords: Protaper, Preclinic, Rotary techniques, Education, Protaper next

Funding: The authors received no funding for this work.

==============================
Background

Dentistry has undergone an evolution in endodontics practice caused by the advancement of rotary techniques for root canal preparation and their subsequent incorporation into the teaching of dentistry undergraduates. This research aimed to evaluate the shaping ability of third-year dental students as their first experience in rotary instrumentation using ProTaper Universal (PTU) and ProTaper Next (PTN) (Dentsply Maillefer) rotary instruments in simulated curved canals.

Methods

Forty students instrumented 200 simulated canals with a 40° curvature in resin blocks according to the manufacturer’s instructions with PTU and 39 students and 195 canals with PTN files. The canals were prepared at a speed of 300 rpm using a 16:1 reduction hand-piece powered by an electric motor (Xsmart; Dentsply Maillefer). The final apical preparation was set to F2 for the PTU and X2 for the PTN group. The change in canal curvature was evaluated based on Schneider technique using the AutoCAD 2007 software on post-digital photographs. The incidence of instrument fracture and deformation, the incidence of ledge, the change in working length (WL), and the working time were noted. The data were analyzed with Student’s t-test and Chi-Square test at a significance level of 0.05 using SPSS.

Results

PTN maintained the original canal curvature better, resulting in fewer fractures and ledges, and shaped the canals faster than the PTU (P < 0.05). The mean curves of the resin canals after the instrumentation for the PTU and PTN groups were 24.03° ± 3.14° and 25.64° ± 2.72°, respectively. Thirty-three (17.4%) PTU and 18 (9.3%) PTN files fractured (p < 0.05). Nine (4.5%) PTU and 2 (2.6%) PTN deformed (p > 0.05). The change in WL after instrumentation was 0.97 mm ± 0.95 mm in PTU and 0.96 mm ± 0.80 mm in PTN (p < 0.05). The mean times were 627 s ± 18 s for PTU and 379 s ± 18 s for PTN (p < 0.000).

Discussion

PTN can be recommended in severely curved root canals in terms of maintenance of the original canal curvature, superior instrument fracture and fewer ledges. Even if training before preparation provides an acceptable level of canal shaping for preclinical students, the use of NiTi rotary instruments should be included in the undergraduate dental curriculum, contributing to an increase in the quality of root canal shaping and, consequently, to an improvement of the clinical experience of students.

Introduction

Cleaning and shaping of the root canal, which must be performed mechanically and biologically, is the most important procedure in root canal treatment. A variety of techniques and instruments have been developed to perform this procedure without generating undesirable clinical results, such as canal straightening, transportation, ledging, strip perforations, or instrument fractures. The European Society of Endodontics (ESE) has published current articles to lead the undergraduate curriculum in endodontics to improve dental students’ theoretical and clinical education (ESE, 2001). The ESE believes that undergraduate students should receive preclinical and clinical training to be able to successfully treat uncomplicated anterior premolar and molar teeth (de Moor et al., 2013). Endodontic education needs to be developed in undergraduate programs (Dummer, 1991; Qualtrough & Dummer, 1997; Unal et al., 2012) and the use of new techniques and devices that have proven to be positive contributions to root canal therapy should be included in undergraduate training programs (Unal et al., 2011).

The technical quality of root fillings, assessed radiographically, performed by undergraduate students using hand instruments was investigated in a meta-analysis (Ribeiro et al., 2018). The results revealed a low frequency (48.75%) of acceptable technical quality of root fillings and drew attention to the major procedural errors created (Ribeiro et al., 2018). Many studies have been conducted to evaluate the performance of students in curved root canals (Hänni et al., 2003; Arbab-Chirani & Vulcain, 2004; Alrahabi, 2015; Unal et al., 2011). Despite the multiple benefits of using NiTi rotary instruments, the step-back technique using stainless steel files is still a conventional teaching method in endodontic programs for undergraduate students in most countries (Alrahabi, 2015). Although students with no experience are reported to be successful in the use of rotary instruments (Unal et al., 2012; Kwak et al., 2016; Brito-Júnior et al., 2014), it is evident that errors can be minimized by the training to be taken in the curriculum. Georgelin-Gurgel et al. (2008) and Alrahabi (2015) reported that manual instrumentation is safer than rotary instrumentation in the hands of inexperienced students. They underlined that acquiring skill in the use of NiTi rotary instrumentation requires specific preclinical training to avert file breakage. Martins et al. (2012) suggested that the use of NiTi rotary instruments should be included in the undergraduate dental curriculum, as this would contribute to an increase in the number of patients assisted and, consequently, improve the students’ clinical experience.

Some procedural errors have been minimized after the introduction of the NiTi alloy into endodontics (Walia, Brantley & Gerstein, 1988; Vaudt et al., 2009; Yang et al., 2011). Nevertheless, the fracture of the NiTi rotary root canal instruments suddenly and without any warning creates a disappointment for the dentist (Zuolo & Walton, 1997; Arens et al., 2003; Ankrum, Hartwell & Truitt, 2004). The studies are underway to increase the resistance of the root canal instrument against breakage in order to overcome this problem. A recent novelty of the alloy is the so-called m-wire alloy. The M-Wire material has been shown to have longer fatigue resistance and lower fracture risk than conventional NiTi alloy does (Johnson et al., 2008; Montenegro-Santillàn et al.,2013).

The ProTaper Universal (PTU) system (Dentsply Maillefer, Ballaigues, Switzerland) is currently used by endodontists, and it is one of the most revolving instrument systems researchers have searched for the most (Peters, Schönenberger & Barbakow, 2003; Schäfer & Vlassis, 2004; Celik Unal, Kececi & Ureyen Kaya, 2006; Yang et al., 2007; Alemam, Dummer & Farnell, 2017). PTU is a nickel titanium (NiTi) rotary system of instruments manufactured with progressive tapering over the length of the cutting blades, convex triangular cross-sections, and noncutting tips. In the PTU system, the file is produced in such a way that it does not cut the ends, and its cross-section is a convex triangle. The taper angle of the file is not constant. This angle increases parabolically starting from the end (Hieawy et al., 2015).

The design features of ProTaper Next (PTN, Dentsply Maillefer), made from the new M-Wire alloy, include variable tapers and rectangle sections with a remote center. The M-Wire material has been shown to have longer fatigue resistance and lower fracture risk than conventional NiTi alloy does (Johnson et al., 2008; Montenegro-Santillàn et al.,2013). The number of instruments is similar to that of the PTU in terms of the order of use and each instrument. The PTU contains three files (SX, S1, and S2) for the preparation of the coronal and middle thirds and three files (F1, F2, and F3) for the preparation of the apical third. PTN consists of only three files (X1 is #17/.04, X2 is #25/.06, and X3 is #30/.075), close to the diameters of the files used in the apical third in the PTU system (Pérez-Higueras et al., 2014).

This study aimed to compare the shaping ability of PTU and PTN rotary instruments used by the third-year dental students inexperienced in any rotary instrumentation techniques in simulated curved canals of resin blocks.

Methods

Seventy-nine third-year undergraduate dental students (2015–2016 intakes) in preclinical endodontics at the School of the Dentistry of the Suleyman Demirel University, Isparta, Turkey had three 50-min lectures of about the NiTi instruments, their physical properties, and the special constructional features of the files. The present study was based on data obtained from practical courses, which was compulsory for the students to attend. Therefore, there was no need to sign an acceptance form for participation. The results of this study did not affect the course notes of the students. The students instrumented 16 simulated canals in acrylic resin blocks in the 2014–2015 academic year and 12 extracted human teeth 2015–2016 in the preclinical dental education. The shaping technique was Modified Double Flared Technique using balanced force principle (Saunders & Saunders, 1992) with stainless steel instrument. The students attended a 2-h lecture on the use of PTU and PTN files two weeks before the end of the term. The study was started in the last week of the term. The students were divided into groups of 10, and an associated professor (G.Ç.) demonstrated the procedures of the shaping of a simulated canal according to the instructions of the manufacturer. A printed script with the manufacturer’s step-by-step instructions for the PTU and PTN rotary systems was given to each student.

The students instrumented a total of 395 simulated resin canals (Endo Training Block 02 taper, REFA 0177; Dentsply Maillefer, CH-1338 Ballaigues, Switzerland). All simulated resin canals had an apical foramen of 0.15 mm, a taper of 0.02, and an angle of curvature of 40°. Of the 79 students, 40 students assigned randomly instrumented 200 canals using PTU files and 39 students instrumented 195 canals using PTN files. Each student instrumented five simulated canals using just one of the two systems and only one set instrument. The canals for each file system were instrumented to a working length of 16 mm (0. five mm from the apex) at a speed of 300 rpm and a torque-control level of 2, using a 16:1 reduction handpiece powered by an electric motor (Xsmart; Dentsply Maillefer). Two mL of distilled water was used as an irrigant at each change of instrument. After the instrumentation, a final rinse with distilled water for 1 min was carried out. Each canal was dried using size 25 paper points.

The incidence of fractures in the blocks and instrument deformations It were recorded. The blocks which instrument fractured were not taken into account when calculating the curvature and WL change, and incidence of the ledge.. The working time was calculated starting through the insertion of the first file until the end of the instrumentation, including total active instrumentation, cleaning of the flutes of the instruments, and irrigation. The final WL of the canals was determined in mm following instrumentation of canals. An F2 PTU or X2 PTN file was inserted into the canal and its WL within the canal measured to the nearest 0. five mm. The amount of change in WL was determined by subtracting the final length from 16 mm.. Canal straightening and ledge formation were assessed on digital images (Figs. 1 and 2). After the canal instrumentation, the images of the resin blocks were obtained with the help of a light microscope camera (Zeiss Axioskop 2; Zeiss, Münich, Germany) to determine the canal curvature. The images were taken from each block in one direction (mimicking clinical conditions). In these images, canal curvatures were measured by a computer program (AutoCAD 2007) according to Schneider (1971). The change in canal curvature was determined by subtracting the value obtained after instrumentation from 40 degrees. All students were randomly divided into two groups as they are at the same educational level. Therefore, it was accepted that the only variable in the experimental groups was PTU and PTN files. Data were analyzed using SPSS software, version 10.0 (SPSS Inc., Chicago, USA). The level of significance was set at 5%. Student’s t-test was used to evaluate differences in canal straightening, working time, and amount of change in WL. The chi-square test was used to evaluate differences in the incidences of instrument fracture, instrument deformation, while the Yates corrected chi-square test was used ledge formation between the groups.

Figure 1 Measurement of the canal curvature on the images captured by stereo microscope after instrumentation.

(A) degree of curvature according to Schneider (1971).

Figure 2 Ledge formation at the beginning of the curvature associated with a marked loss of working length.

The arrow indicates the ledge.

Results

The PTN preserved the original canal curvature (P < 0.000) better, resulting in fewer fractures (p < 0.05) and ledges (p < 0.000), and shaped the canals faster (P < 0.000) than the PTU. The mean curvatures of the canals after instrumentation with PTU and PTN were 24.03° ± 3.14° and 25.64° ± 2.72° , respectively (P < 0.000). The incidence of instrument fracture was 17.4% (33) for PTU and 9.32% (18) for PTN (p < 0.05). The incidence of instrument deformation was 9 (4.5%) for PTU and 2 (2.6%) for PTN (P?0.05). The PTN had fewer ledges than PTU (P < 0.000). Ledge formation was observed in 33 (18.8% () of the canals in the PTU group, while in 7 (4%) in the PTN group (P < 0.000). There was no statistically significant difference in the preservation of the WL (P > 0.05). The mean change in WL after instrumentation was 0.97 mm ± 0.95 mm in the PTU group and 0.96 ± 0.80 mm in the PTN group (p > 0.05) PTN shaped the canals more quickly than PTU (P < 0.000). The mean time in instrumentation of the canals was 627 s ± 28 s for PTU and 379s ± 18s for PTN (P < 0.000) (Table 1).

Table 1 The canal curvature, the change in WL, the working time, the fractured instrument and deformed instrument, and the ledge based on the groups after instrumentation.

	PTU	N	PTN	N	total	P value	
The degree and SD of canal curvature	24.03° ± 3.14°	174	25.64° ± 2.72°	176	395	0.000∗∗∗	
Amount of change and SD in WL (mm)	0.97 ± 0.95	155	0.96 ± 0.80	174	395	0.99	
Working time and SD (s)	627 ± 28	200	379 ± 18	195	395	0.000	
The number and percent of the canal with fractured instrument	17.4%
(n = 33)	190	9.32%
(n = 18)	194	395	0.020∗	
The percent of deformed instrument	4.5%
n = 9	200	2.6%
n = 2	78	278	0.403	
The percent of ledge	18.8%
n = 33	176	4%
n = 7	176	352	0.000∗∗∗	
Notes.

SD standard deviation

PTU ProTaper Universal

PTN Protaper Next

N the number of sample

Discussion

The ESE, Education and Scholarship Committee, encourages the use of proven new tools, techniques, and training resources in endodontics education (de Moor et al., 2013) Updated of the endodontic clinical curriculum is an essential step to having acceptable endodontic treatment, for both general practitioners and specialist dentists. Preclinical exercises are extremely important for the acquisition of this skill (de Moor et al., 2013; Georgelin-Gurgel et al., 2008; Alrahabi, 2015).

This research presents the experiences of the third-year preclinical students in the Faculty of Dentistry of the University of Suleyman Demirel with rotary instruments following training on root canal preparation with hand instruments. Instrumentation techniques with engine-driven instruments are now almost indispensable for root canal treatment beyond popularization in root canal instrumentation. Along with hand instruments, which are indispensable, instrumentation with rotary instruments is also important. Hänni et al. (2003) showed that third-year students can use preclinical course profile 0.04 taper (Dentsply Maillefer, Ballagues, Switzerland) instruments in a satisfactory manner. Moreover, Arbab-Chirani & Vulcain (2004) in France and Abu-Tahun et al. (2014) in Jordan reported that there is a consensus on the need for the use of rotary instruments in undergraduate education. Following this, the use of rotary instruments in the dentistry faculty was added to the curriculum in France. Kang et al. (2006) reported that the ProFile is the safest, best instrument for root canal shaping for students and beginners. Peru et al. (2006) reported that curved root canals prepared by non-experienced undergraduates with rotary instruments exhibit less procedural errors and require less time than those generated using hand instruments. In addition, Tu et al. (2008) showed that undergraduate students could generate more successful root canal shaping with ProTaper rotary instruments than they can with hand ProTaper instruments. Sonntag et al. (2008) found that preclinical endodontics training varied significantly according to the curriculum, instructor, and course content, and they announced that, as of 2008, most (63%) faculties in Germany taught root canal instrumentation with rotary Ni-Ti instruments. Unal et al. (2012) reported that third-year students with no experience with rotating instruments were successful in molar root canal preparations after 2 h of theoretical education about rotary instruments. Brito-Júnior et al. (2014) reported that undergraduate students produced lower apical transportation in curved canals with F1 and F2 PTU files. Furthermore, Bruno et al. (2016) stated that the low fracture rates observed in their study indicated that the examined instruments can be used safely by students.

In the literature, there are many different experiments to evaluate the shaping and cleaning of the root canal. Artificial resin models are preferred in many studies because having a standard canal length, curvature, and form (Kum et al., 2000; Unal et al., 2009; Unal et al., 2012). The CT is emerging in several endodontic research facilities as a nondestructive and accurate method to analyze canal geometry and the relative effects of shaping techniques (Peters, Schönenberger & Barbakow, 2003; Peru et al., 2006). As the cost of the CT assessment method was high, our study was carried out in acrylic resin blocks. Although this experimental model does not fully reflect the morphology of real human teeth, it can give an idea of relation to relation to the performance of the root canal instrument.

The most frequently encountered case for procedural errors during the preparation of curved root canals is the straightening of the root canals instrument fracture and deformation, and ledge (Nagy et al., 1997). Some of the reasons for these procedural errors are the instrument type and dimension, the type of alloy and the canal curve of before the instrumentation (Parashos & Messer, 2006; Lambrianidis, 2009). Further, the test material in which the instrumentation is carried out may also affect the test results (Alrahabi & Zafar, 2018). In our study, PTN (25.64∘) maintained original canal curvature better than PTU (24.03∘) (P < 0.000). The amount of straightening was found to be higher compared with Unal et al., 2012’s (2012) study. The researchers selected the F1 PTU file as the master apical file. The smaller the size of this file than the F2 PTU file, the less straightening it may have caused. Besides, the shaping ability of the files was examined in acrylic resin blocks in our study, while Unal et al. (2012) tested in extracted human teeth. Alrahabi & Zafar (2018) reported that the files show different performance in the extracted human teeth and simulated resin canals.

The second of the most frustrating procedural errors encountered during the instrumentation of curved root canals is the abrupt fracturing and deforming of the instruments. Applying enough force to metal causes the shape of the material to change, and this shape change is called deformation. After the force is removed, the permanent shape change that cannot be returned to its original state is called plastic deformation (Thompson, 2000). The opening in the grooves manifests the permanent deformation. In this study, deformation was detected under a microscope. The incidence rate of deformation was 4.5% (9) for PTU and 2.6% (2) for PTN (p > 0.05).

The incidence rates of fracture were 17.4% for PTU and 9.3% for PTN (p < 0.05) in our study. Vieira et al. (2008) reported that unexpected instrument fractures in the automatic systems were closely related to the operator’s experience. On contrary, it was reported that inexperienced students could shape the root canal as satisfactory (Hänni et al., 2003; Peru et al., 2006; Unal et al., 2012; Brito-Júnior et al., 2014). Bruno et al. (2016) announced that a total of 30 file fractures were noted during the study period; thus, fractures occurred in 0.37% of total file uses and 2.98% of all works. The authors attributed a low rate of fractures to some factors. These factors are the use of the F3 ProTaper, the experience of the operator, and supervised treatment by expert endodontists. Also, Bruno et al. (2016) designed this study under clinical circumstances. Therefore, since the study included all tooth types, the average curvature of the teeth could be assumed to be less than 40 degrees. Besides, working on more curved canals and acrylic resin blocks instead of real teeth could be increased the numbers of instrument fractures and deformations. One of the disadvantages of the use of rotary canal instruments in resin blocks is that they heat the resin while the instruments are in operation, and this heat causes the resin to soften. This heat can also cause the canal instrument to bind to and become entangled in the softened resin, thereby breaking the instrument (Baumann & Roth, 1999; Kum et al., 2000). The choice of acrylic resin blocks for standardization is also a limitation of this study. Nevertheless, in a study by Alemam, Dummer & Farnell (2017) comparing the shaping abilities of the PTU and PTN instruments in S-shaped canals, the students were more successful. This difference may be due to their use of aqueous lubricant during the preparation. Lubrication usually helps prevent fracture of the instrument during root canal preparation by helping the instrument work in the canal without being forced (Zehnder, 2006). We did not use lubricants in our study; we only used distilled water for irrigation. An aqueous lubricant has been shown to be more effective than a gel-type lubricant. A lubricant may exert a physical effect by floating debris away from the rotating instrument. In addition, chemical additives may act on the root canal dentin to facilitate instrumentation (Peters, Boessler & Zehnder, 2005; Boessler, Peters & Zehnder, 2007). In our study, this difference between PTU and PTN in terms of instrument fracture and instrument deformation can be due to the different levels of flexibility of the instruments. While PTU is produced from a conventional NiTi alloy, PTN is produced from the more flexible M-Wire. Furthermore, The PTN instruments have an innovative off centered rectangular cross-section that gives the file a snake-like swaggering movement as it progresses through the root canal. The manufacturer claims that this asymmetric rotary motion of ProTaper Next allows achieving fully tapered canals with fewer numbers of files (Tulsa Dental Specialties, 2017). In addition, the X2 of PTN has an apical cone of 0.06, while PTU’s F2 instrument exhibits a constant taper from D1 to D3 (0.08) (Alemam, Dummer & Farnell, 2017).

A ledge is defined as a deviation from the original canal curvature within the apical third which creates or starts to create a new canal at a tangent to the original canal (Nagy et al., 1997). In our study, ledges occurred in 19% of the canals prepared with PTU, whereas ledges formed in 4% of PTN cases. One of the reasons why PTN files produce fewer ledges than PTU files can be the superior properties of the m-wire alloy. The alloy structure and cross-sectional shape of the PTU system’s F1 and F2 instruments will result in a hardened instrument that can cause more ledges to occur than arise in PTN. Furthermore, Alrahabi & Zafar (2018) considered contributing to the snake-like, swaggering movements of the files during advancement into the root canal. This movement serves to minimize the engagement between the file and dentin. Reduced engagement limits any undesirable taper lock, the screw effect, and the torque on any given file (Haapasalo & Shen, 2013).

Conclusions

Within the limitations of this study, the PTN can be preferred in severly curved root canals in terms of maintenance of the original canal curvature, superior fracture, and fewer ledges. Even if short training before preparation provides an acceptable level of canal shaping for preclinical students, the use of NiTi rotary instruments should be included in the undergraduate dental curriculum, contributing to an increase in the quality of root canal shaping, and consequently the improvement of clinical experience of students.

Supplemental Information

Dataset S1 Raw data

Each data point indicates the performance of the groups.

Click here for additional data file.

The authors thank Dentsply® for the root canal files and resin blocks used in this study.

Additional Information and Declarations

Competing Interests

Author Contributions

Data Availability

Feyza Özdemir Kısacık is an employee of Vefa Oral and Tooth Health Center. Emir Yilmaz is an employee of Dentplus Oral and Dental Health Center.

Gül Çelik conceived and designed the experiments, performed the experiments, analyzed the data, prepared figures and/or tables, authored or reviewed drafts of the paper, approved the final draft.

Feyza Özdemir Kısacık, Emir Faruk Yılmaz, Arife Mersinlioğlu and İhsan Furkan Ertuğrul performed the experiments, contributed reagents/materials/analysis tools.

Hikmet Orhan analyzed the data.

The following information was supplied regarding data availability:

The raw measurements are available as Dataset S1.

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
