# Peer review of "A comparative study of root canal shaping using protaper universal and protaper next rotary files in preclinical dental education"

_PeerJ, doi:10.7717/peerj.7419_

## Round 0.1 · original submission · Major Revisions

Thank you for submitting your work to our Journal. However, you need to adress all the comments of the reviewers carefully before resubmitting your manuscript.

Reviewer 1 ·

Basic reporting

• There is unclear, ambiguous language in some sections.
• What is/are statistical test(s) that was/ were used in this study? should be mentioned in background, methods, and results
• What is deformation? Need to be defined
• Figures should be included (shaping ability study)

Experimental design

The two different systems (PTU and PTN) were used by different groups of students and this hinder the power of this study, it is indirect comparison with unnecessary variables.

Validity of the findings

More information about data collection and statistical analysis is needed

Additional comments

• There is unclear, ambiguous language in some sections.
• What is/are statistical test(s) that was/ were used in this study? should be mentioned in background, methods, and results
• The two different systems (PTU and PTN) were used by different groups of students and this hinder the power of this study, it is indirect comparison with unnecessary variables.
• What is deformation? Need to be defined
• Figures should be included (shaping ability study)

Annotated reviews are not available for download in order to protect the identity of reviewers who chose to remain anonymous.

Reviewer 2 ·

Basic reporting

No comment

Experimental design

No comment

Validity of the findings

No comment

Additional comments

The study was well designed and professional english language was used (Certificate of editing and proofreading was attached). Research question well defined, relevant & meaningful. The first author has experience in this field.
Even so, the authors must be controlled the references. (for example; authors name in the text ; in a study by Alemann et al. (2017), and in the references ; Alemam AAH, Dummer PMH, Farnell DJJ. 2017, are different.)
As a result, this article is acceptable after correction of subject above.

Reviewer 3 ·

Basic reporting

The author should mention the importance of using NiTi rotary instrumentation by the undergraduate dental student with more details, the author can refer to this study:
Comparative study of root-canal shaping with stainless steel and rotary NiTi files performed by preclinical dental students
Technology and Health Care 23 (3), 257-265

Experimental design

Was the participation of students in this study mandatory or voluntary?
is there any acceptance form for participation in this study?
is there any relation between students grades in this course and participating in this study
All these points should be mentioned in the materials and method section

Validity of the findings

The findings are valid

Additional comments

The manuscript provides a good idea but needs several modifications to be more valuable work where I have the following comments;
Introduction section:
1. The author should mention the importance of using NiTi rotary instrumentation by the undergraduate dental student with more details and the author can refer to this study:
Comparative study of root-canal shaping with stainless steel and rotary NiTi files performed by preclinical dental students
Technology and Health Care 23 (3), 257-265
Methods section
Was the participation of students in this study mandatory or voluntary?
is there any acceptance form for participation in this study?
is there any relation between students grades in this course and participating in this study
All these points should be mentioned in the materials and method section

The students had applied root canal treatment to a total of 12 extracted human teeth in the phantom laboratory
what is the instrumentation technique do the student used stainless steel file with step back technique

Canal straightening and the incidence of ledges were assessed on digital images with AutoCAD 2007.
This part needs more expansion

Discussion section
Narayanaraopeta & AlShwaimi (2015) announced that five of the six schools employed
NiTi rotary instruments. Please this statement needs more details and clarification
Muñoz et al. (2014) showed that students also engaged in satisfactory shaping via reciprocal movement with a single file. but in your study, you used rotary systems. and you could improve your study by comparing a rotary system with a reciprocating single file system

These researchers assessed the amount of straightening of the channels prepared by the third-year students, who had no experience with the rotary instruments, in the extracted molar teeth.
Is there no effect of the difference between dentin in the root canal and resin in simulated root canals on the apical transportation you can refer to the following article:
Assessment of apical transportation caused by nickel-titanium rotary systems with full rotation and reciprocating movements using extracted teeth and resin blocks with simulated root canals: A comparative study
In severely curved canals, the master apical size should be smaller than #30; Where is the reference?
In our study, the incidence rates of deformation and fracture were 17% for PTU and 9% for PTN. Can you explain this difference in fracture rate between PTU and PTN

Ledges occurred in 19% of the canals prepared with PTU, whereas ledges formed
in 4% of PTN cases. Does swaggering motion in PTN has any effects in reducing ledges

Within the limitations of this study, What are the limitations of your study?
and what is your suggestion to improve teaching and using NiTi systems by undergraduate dental students

Annotated reviews are not available for download in order to protect the identity of reviewers who chose to remain anonymous.

---

## Round 0.2 · accepted · Accept

The authors fulfilled all reviewers comments

Reviewer 2 ·

Basic reporting

No comment

Experimental design

No comment

Validity of the findings

No comment

Reviewer 3 ·

Basic reporting

The manuscript is written in clear English in professional method, with sufficient background and referenced appropriately.
The article structured in an acceptable format .
Figures relevant to the content of the article, of sufficient resolution, and appropriately described and labeled.
The manuscript is self-contained with relevant results to hypotheses.

Experimental design

The manuscript has a clear define research question, with high technical standard.
Methods described with sufficient information and can be reproducible by another investigator.

Validity of the findings

The findings of this manuscript are rationale & benefit to literature .
The conclusions are well stated, linked to original research question and supported by results.

Additional comments

this manuscript could be considered suitable for publication now and provide valuiabl information in field of teaching endodontic